# Protein Fortification of Millet-Based Gluten-Free Snacks Designed for 3D Printing

**DOI:** 10.3390/foods14244308

**Published:** 2025-12-14

**Authors:** Jovana Simeunović, Jelena Miljanić, Bojana Kokić, Lidija Perović, Jelena Jovančević, Jovana Glušac, Jovana Kojić

**Affiliations:** 1Institute of Food Technology in Novi Sad, University of Novi Sad, Bulevar Cara Lazara 1, 21000 Novi Sad, Serbia; jovana.simeunovic@fins.uns.ac.rs (J.S.); jelena.miljanic@fins.uns.ac.rs (J.M.); bojana.kokic@fins.uns.ac.rs (B.K.); lidija.perovic@fins.uns.ac.rs (L.P.); 2BioSense Institute—Research and Development Institute for Information Technologies in Biosystems, University of Novi Sad, Zorana Đinđića 1a, 21000 Novi Sad, Serbia; jelena.jovancevic@biosense.rs (J.J.); jovana.glusac@biosense.rs (J.G.)

**Keywords:** gluten-free ready-to-eat snack, 3D-printed snack, yeast protein, almond protein, natural foods

## Abstract

The global trend in gluten-free snack innovation involves using naturally gluten-free grains as a nutrient-rich foundation, enriching formulations with multifunctional plant and microbial proteins, and optimizing ingredient interactions to balance nutritional demands with structural integrity. The study demonstrates that blending proso millet flour with yeast-derived and almond (50:50 ratio) proteins effectively produces a protein- and fiber-rich gluten-free dough suitable for 3D printing, without the need for synthetic additives. This approach aligns with the growing demand for clean-label, sustainable protein sources that enable the creation of healthy, stable, and appealing ready-to-eat snacks. The enriched dough demonstrated superior rheological behavior, characterized by a dominant elastic modulus (G′ > G″), enabling smooth extrusion and stable shape retention. Nutritional analysis revealed an increase in protein (28.16 vs. 13.26 g/100 g DB) and dietary fiber (12.17 vs. 6.22 g/100 g DB) compared to the control. The amino acid profile improved significantly, with 48% more essential amino acids and a 63% increase in non-essential amino acids. Dimensional accuracy improved, and post-processing deformation was reduced, confirming enhanced structural integrity. Texture analysis showed no significant increase in hardness, maintaining a desirable texture profile despite higher protein content. Sensory evaluation confirmed greater acceptance of the enriched snack, especially in terms of flavor, aroma, and smell, while preliminary cost assessment indicated that, despite higher ingredient costs, the enriched formulation remains economically feasible. Additional optimization of protein concentration and processing conditions could enhance the overall functionality even further.

## 1. Introduction

The increasing prevalence of celiac disease and gluten sensitivity, combined with a growing consumer shift toward health-conscious and wellness-oriented diets, has fueled substantial demand for gluten-free food products [1]. In particular, gluten-free snacks are gaining considerable attention as convenient and nutritious options [2]. However, developing gluten-free snacks that are both nutritionally rich and structurally stable presents a challenge. This challenge becomes even more acute with innovative manufacturing techniques such as 3D food printing, which requires precise control of dough rheology and shape retention during and after production [3,4].

In 3D food printing, maintaining shape fidelity and texture requires harmonizing ingredient interactions with optimized printing parameters such as nozzle diameter, extrusion speed, and layer height [5]. This technological tuning, combined with ingredient innovations, enables the creation of gluten-free snacks that meet both consumer expectations for nutrition and manufacturers’ quality control demands.

Gluten plays a critical role in conventional wheat-based doughs by providing viscoelastic properties that contribute to elasticity, extensibility, and structural stability [6]. Naturally gluten-free grains, such as millet, sorghum, rice, buckwheat, and amaranth, are widely studied as wheat substitutes to meet gluten-free requirements while providing essential nutrients such as dietary fiber, vitamins, and minerals [7]. Among these, proso millet flour has attracted attention due to its high content of fiber, calcium, and B vitamins [8]. Nevertheless, millet-based doughs lack gluten’s characteristic viscoelasticity, resulting in doughs that are difficult to shape and maintain during food processing, particularly in 3D printing. This deficit significantly impairs the ability to create stable, well-defined snack structures [4,9].

Besides gluten absence, modern consumers increasingly demand higher protein content in snacks, motivated by health trends emphasizing satiety, muscle maintenance, and plant-based diets [10]. However, the addition of high protein levels often introduces new formulation challenges. High-protein doughs tend to become less cohesive, more brittle, or overly dense, negatively impacting printability and structural integrity [11]. Thus, successfully combining gluten-free formulations with protein enrichment without sacrificing processability or finished product quality remains a critical hurdle.

Multiple studies address the challenge of balancing nutrients and texture by blending proteins from different sources [12]. Recently, Krishnaraj et al. [13] and Hussain et al. [14] reflected several approaches toward this dual objective of nutrition and printability in gluten-free snacks. Researchers have reported the successful use of composite flours combining millet with legumes like green gram and fried gram for acceptable 3D-printed snack products rich in fiber and protein. Plant protein isolates, such as pea protein, have also proven effective in fulfilling daily reference protein intake targets while maintaining dough properties conducive to 3D printing [14].

Yeast protein, a sustainable and underutilized single-cell protein source, offers a complete essential amino acid profile comparable to high-quality animal proteins [15]. It possesses advantageous techno-functional properties, such as emulsification and water-binding, which support dough structure [15]. While yeast protein has been investigated in gel-based 3D-printed foods [16], to date, no studies have investigated its application specifically in 3D-printed snack formulations. Thus, its application within solid snack formulations represents a novel approach with promising potential for sustainability and clean-label claims. Almond protein complements this blend with a mild, consumer-friendly flavor and significant techno-functional benefits, including water and oil retention capacities [17]. Nutritionally, almond protein is rich in branched-chain amino acids, arginine, and micronutrients such as magnesium and vitamin E, contributing not only to enhanced protein content but also to improved texture and dough handling [17].

To overcome these intertwined challenges, this study explores the innovative use of a protein blend comprising yeast-derived protein (YP) and almond protein, incorporated with proso millet flour, to develop 3D-printable gluten-free dough with enhanced nutritional and functional qualities. Moreover, the complementary properties of combining protein sources (yeast-based and nut-based) demonstrate synergistic effects, enhancing dough elasticity and strength. This natural ingredient synergy contrasts favorably with strategies relying on additives like xanthan gum, guar gum, psyllium husk, or pectin, supporting an increasing consumer preference for minimally processed, natural foods.

## 2. Materials and Methods

### 2.1. Materials

*Saccharomyces cerevisiae* (yeast) protein cookie flavor (75.87% proteins, 9.14% fat, 6.40% dietary fibers) under the commercial name Smartein was kindly donated by Pharma Intelligence, Novi Sad, Serbia. The proso millet flour (10.15% proteins, 2.96% fat, 3.90% dietary fibers) (Interpak, Kraljevo, Serbia), almond protein (46.2% proteins, 15.5% fat) (Crazy Nutrition, Belgrade, Serbia), and almond butter with cinnamon (20.18% proteins, 48.89% fat) (Granum, Hajdukovo, Serbia) were purchased from the local supermarket in Novi Sad, Serbia.

### 2.2. Ink Preparation

The experimental plan was designed to determine the optimal protein formulation for producing structurally stable and nutritionally enriched 3D-printed millet-based snacks. The experiment was conducted in two sequential stages.

The control dough formulation consisted of millet flour, almond butter, and water. In the first stage, a portion of the flour was replaced by protein powders (50:50 ratio) to obtain protein-enriched doughs (containing 12%, 16%, and 20% total protein, calculated on a wet basis). The objective was to identify the protein level that provides an optimal balance between nutritional enhancement and printability. Accordingly, the experimental design focused on assessing the influence of incremental protein incorporation on key printing performance parameters, shape fidelity, post-processing deformation, texture, and appearance, which were selected as critical indicators of overall print quality and structural stability.

In the second stage, the total protein level was fixed at 16%, corresponding to the formulation that exhibited the most favorable performance in the preliminary trials. At this protein level, the ratio between almond protein and YP was systematically varied to assess the effect of each source and their interaction on dough handling and printing performance. The tested ratios were 100:0, 0:100, 75:25, 25:75, and 50:50 (*w*/*w*), as shown in Table 1.

The obtained data were used to rank all formulations according to the defined performance criteria, and the formulation with the most favorable combination of structural and compositional attributes was selected as the optimal protein ratio for subsequent analyses.

During ink preparation, all fractions were passed through a 0.5 mm sieve to ensure uniform particle size distribution. Almond butter with cinnamon was tempered at room temperature (25 °C) and added to the dry ingredients. The mixture was initially mixed manually with a spatula in a circular motion. After the addition of water, the mixture was homogenized for 1 min at 300 rpm using a hand mixer (MX2002, VOX, Belgrade, Serbia) to form cohesive dough and left to rest for 30 min at room temperature.

### 2.3. Rheological Measurements

The rheological properties were examined utilizing a dynamic rotational and oscillatory rheometer (HAAKE Mars, Thermo Scientific, Dreieich, Germany) with a parallel plate (Type PP35S, 1 mm gap) at 25 °C. An amplitude sweep test was conducted at 1 Hz to determine the total linear viscoelastic (LVE) range between stress 0.1% and 1000%. The frequency sweep test was determined to evaluate elastic modulus (G′, Pa), viscous modulus (G″, Pa), loss factor (tanδ = G″/G′), and complex viscosity (η*, Pas). This test was performed at 1–10 Hz as per the established LVE-R and stress recommended by the previous test. The shear-viscosity test was examined in a ramp linear mode, with the shear rate varying from 0.01 to 100 s^−1^ at a temperature of 25 °C. The measured values of shear strain and stress were fitted to the Power law model (η = Kγ^(n−1)^), where η stands for shear stress (Pa), γ indicates the shear rate, K refers to the consistency index, and n represents the flow behavior index. All measurements were conducted in duplicate and performed at 25 °C.

### 2.4. Three-Dimensional Printing Process

Three samples of each probe were printed, using a 3D Food printer (Felix Food 1.5, Felix Printers, IJsselstein, The Netherlands), into a butterfly shape (dimensions: 60 mm × 60 mm × 6 mm). Computer-aided design (CAD) of a butterfly was designed in Autodesk 3ds Max 2022.2 (Autodesk Incorporation, San Francisco, CA, USA). The butterfly shape was saved as a stereolithography (.stl) extension file and sliced in the Simplify3D software version 5.1.2. The extrusion system of the 3D printer is constructed of a 100 mL capacity syringe with retractable plunger and stainless-steel nozzle. Printing parameters were set as follows: Infill percentage = 30%; Extrusion multiplier = 120%; Printing speed = 25 mm/s; Nozzle diameter = 1.60 mm (14G); Layer height = 90% of nozzle diameter; Infill type = grid; Temperature = 35 °C.

### 2.5. Post-Processing

Post-processing of the printed 3D snacks was performed using a drying oven with air flow (Sterimatic ST-11, Instrumentaria, Zagreb, Croatia), at 150 °C until constant mass of the 3D snacks. Oven was preheated at 150 ± 1 °C for 1 h. According to the time required to reach constant moisture within the snack below 5% [18,19], duration of baking was determined through preliminary research and was set for 11 min for each snack. Moisture content was determined using a fast moisture analyzer (model HE53, Mettler Toledo, Columbus, OH, USA).

Mass of the raw and baked snacks was measured on a technical balance (BWL 61, Boeco, Hamburg, Germany). Additionally, baking loss was calculated using the following equation:(1)$$\mathrm{Baking}\;\mathrm{loss}\;(\%)\;=\;\frac{m_{0}-m_{1}}{m_{0}}\times 100,$$
where *m*_0_ is the weight of a 3D snack after printing and *m*_1_ is the weight of a 3D snack after post-processing.

### 2.6. Print and End-Product Quality

An optical 3D scanner (Transcan C, Shining 3D, Hangzhou, China) equipped with EXScan C software version 1.4.2.1 was used for scanning 3D-printed snacks in order to examine print quality. The geometrical measurements were obtained in Siemens Solid Edge 2022 software version 222.00.00132. To evaluate print and end-product quality, responses such as shape fidelity and deformation were calculated using following equations:(2)$$\mathrm{Shape}\;\mathrm{fidelity}\;(\%)\;=\;\frac{X_{1}}{X_{0}}\times 100,$$
where *X*_0_ is the dimension of the 3D CAD model projected in Simplify3D software and *X*_1_ is the dimension of the 3D snack after printing, and:(3)$$\mathrm{Deformation}\;(\%)\;=\;\frac{X_{0\;\;}-X_{1\;\;}}{X_{0}}\times \;100,$$
where *X*_0_ is the dimension of the 3D snack after printing, and *X*_1_ is the dimension of the 3D snack after baking.

3D snacks were frozen at −20 °C for 30 min before the scanning process to maintain the snack shape unchanged. After baking, all snack samples were left for 3 h at room temperature prior to following analysis.

### 2.7. Chemical Analysis

Moisture, ash, fat (AOAC, 2006) [20], protein (ISO 1871:2009) [21] and dietary fiber (991.43, AOAC 2000) [22] content were determined in control and protein-enriched 3D snacks. Total carbohydrate content was calculated using following equation [23]:
Carbohydrate (%) = 100% − (moisture + ash + proteins + fats) %
(4)


### 2.8. Amino Acid Profile

The amino acid composition was analyzed using ion-exchange chromatography on a Biochrom 30+ automatic amino acid analyzer (Biochrom, Cambridge, UK), performed following the official method [24] at the Institute of Food Technology (FINS) of the University of Novi Sad, Serbia [25]. This method involves the separation of amino acids via strong cation-exchange chromatography, followed by reaction with ninhydrin and photometric detection at 570 nm, except for proline, which was detected at 440 nm. L-norleucine (Sigma-Aldrich, St. Louis, MO, USA) served as an internal standard in each analysis. Samples were hydrolyzed in 6 M HCl (Merck, Darmstadt, Germany) at 110 °C for 24 h. For tryptophan determination, alkaline hydrolysis with 4 M NaOH was applied. After hydrolysis, the samples were cooled to room temperature and dissolved in 25 mL of loading buffer (pH 2.2) (Biochrom, Cambridge, UK). They were then filtered through a 0.22 µm PTFE membrane filter (Merck KGaA, Darmstadt, Germany), transferred to vials (Agilent Technologies, Santa Clara, CA, USA), and refrigerated until analysis. Amino acid peaks were identified by comparing retention times with those of a standard amino acid solution (Sigma-Aldrich, St. Louis, MO, USA). Results were expressed either on a total mass of snacks (g/100 g sample).

Using the obtained data (expressed as g/100 g of sample), an indicative Amino Acid Score (AAS) was calculated after conversion to mg per g of protein based on measured protein contents. The AAS was determined by dividing the measured amino acid values (mg/g protein) by the FAO/WHO/UNU [26] reference values for adults (>18 years) [27].

### 2.9. Fatty Acid Profile

Total lipids were extracted using a chloroform/methanol mixture (2:1, *v*/*v*) under continuous agitation for 2.5 h at room temperature. The obtained extract was filtered through Whatman filter paper and subsequently evaporated to dryness. Fatty acid methyl esters (FAMEs) were then prepared via trans-methylation with a 14% boron trifluoride methanol solution, following the ISO 12966-2 method [28].

The prepared FAMEs were analyzed using capillary gas chromatography on an Agilent 7890A system (Agilent Technologies, Santa Clara, CA, USA) equipped with a flame ionization detector (FID) and an SP-2560 fused-silica capillary column (100 m × 0.25 mm × 0.20 µm film thickness). The Supelco 37 FAME mix (Supelco, Bellefonte, PA, USA), containing 37 fatty acid methyl esters, was used as an internal standard for sample analysis. Results were reported on a total fat basis (g/100 g fats).

### 2.10. Texture Analysis

The hardness of 3D-printed snacks was evaluated using a TA.XTPlus Texture Analyzer (Stable Micro Systems, Godalming, UK) equipped with a three-point bending rig (HDP/3PB). Instrumental settings were adapted from the project template BIS2_KB, based on standard settings available in the Texture Exponent Software (version 6.1.6.0, Stable Micro Systems), utilizing a 50 kg load cell. Hardness was defined as the maximum force required for breaking the snack into two parts [29]. Measurements were performed on five individual snack samples.

### 2.11. Sensory Analysis

In total, 40 consumers participated in the study, comprising 26 females and 14 males, with ages ranging from 18 to 65 years. All participants received detailed information about the study and signed informed consent forms before the testing began. The 7-point hedonistic sensory evaluation was carried out through an online questionnaire, with further details provided in the Appendix A. During the sensory session, panelists were asked to assess, on a 7-point Likert scale from “Dislike very much” (1) to “Like very much” (7), the sensory attributes (color, appearance, aroma, texture, flavor, crispiness, and smell). To assess the overall acceptance of the samples, participants were instructed to evaluate each sample on a 5-point scale, where 1 represented the lowest level of acceptance and 5 represented the highest level of acceptance. In addition, re-consumption intention was evaluated by asking panelists whether they would consume each sample again (Yes/No). Given that the data did not conform to the assumptions of normality required for parametric comparisons, differences between the control and enriched samples were assessed through the application of the non-parametric Mann–Whitney U test. Statistical significance was evaluated at a threshold of *p* < 0.05, with values below this cut-off considered indicative of meaningful differences between the groups.

### 2.12. Color and Water Activity Measurements

The color of post-processed snacks was measured using Minolta Chromameter (Model CR-400, Minolta Co., Osaka, Japan) 24 h after post-processing. The results were expressed in terms of L* (lightness), a* (redness to greenness) and b* (yellowness to blueness). Total color difference was calculated to identify color change between post-processed control and enriched snacks:(5)$$∆E=\sqrt{{∆L*}^{2}+\;{∆a*}^{2}+{∆b*}^{2}}$$
where Δ*L** represents difference in lightness, Δ*a** represents difference in redness, and Δ*b** represents the difference in yellowness between samples. To ensure consistency, the snacks were homogenized by grinding, and measurements were conducted in triplicate.

Water activity (a_w_) of post-processed snacks was determined using LabSwift-aw meter (Novasina AG, Lachen, Switzerland). Measurements were repeated three times for both control and enriched post-processed snacks.

### 2.13. Preliminary Laboratory-Scale Cost Benchmarking

A preliminary laboratory-scale cost benchmarking was conducted to provide an illustrative, order-of-magnitude comparison between the control and protein-enriched 3D-printed snack formulations under experimental conditions only. The calculation included the costs of raw materials, electricity consumption during 3D printing and baking, and personnel inputs. Retail ingredient prices were obtained from European online markets, while electricity costs were based on the average late-2024 retail rate of 0.145€ per kWh. Personnel costs were calculated according to standard laboratory hourly rates. The cost per 100 g mixture and per printed snack was estimated for both the control and enriched formulations. Costs related to equipment amortization, industrial labor, packaging, logistics, quality control, certification, and regulatory compliance were not included, as these fall outside the scope of laboratory-scale experimentation.

### 2.14. Statistical Analysis

The data obtained from the experiments were presented as mean value ± standard deviations and statistically analyzed by One-way Analysis of Variance (ANOVA). Significant differences between the means were observed by the Tukey HSD test (*p* < 0.05) using the software Statistica 14.0.0.15 and Python 3.x (Google Colab environment).

## 3. Results and Discussion

### 3.1. Effect of Protein Content and Composition

The increase in total protein concentration from 0% to 20% significantly affected both printing and structural performance of 3D-printed millet-based snacks (Figure 1). Detailed physical and dimensional characteristics of protein-enriched snacks are provided in Appendix A. Consistently, one-way ANOVA detected significant between-group effects for shape fidelity, deformation, and hardness (all *p* < 0.001; Appendix A), and Tukey’s HSD localized the principal contrasts.

The control sample exhibited limited structural integrity and inferior shape fidelity (length = 105.30%, width = 109.01%, height = 77.78%), reflecting the absence of a continuous protein network necessary for maintaining the printed geometry. The 12% protein formulation displayed good flowability during extrusion but still exhibited noticeable spreading and loss of shape fidelity (length = 104.54%, width = 106.41%, height = 90.50%), suggesting insufficient structural reinforcement. Conversely, the dough with 20% protein exhibited significantly reduced shape fidelity (length = 98.84%, width = 93.07%, height = 94.29%) and increased deformation values (length = 6.56%, width = 4.17%, height = 3.91%) compared to the other tested formulations. The ink containing 16% protein achieved the most balanced performance, combining good extrudability, shape fidelity (length = 103.29%, width = 105.06%, height = 96.83%), and minimal deformation after baking (length = 4.34%, width = 2.31%, height = 2.98%).

In terms of texture, a clear and progressive trend was observed (Figure 1). The hardness of snacks exhibited a gradual rise with the incorporation of protein, increasing from 12.13 N in the control formulation to 19.84 N with 20% protein enrichment. This observation is consistent with previous findings showing that protein enrichment generally increases the hardness of printed food structures [30,31,32]. The increase in hardness at higher protein levels can be attributed to intensified protein network formation within the dough matrix, leading to the formation of a denser structure and reduced flowability [33]. This change in mechanical behavior was well reflected in the geometrical accuracy of the printed samples. As protein concentration increased, a denser protein network was formed, improving shape fidelity and structural characteristics. However, beyond 16%, the elevated stiffness limited material flow and layer adhesion, directly affecting shape fidelity and deformation characteristics of 20% enriched snack. Collectively, the ANOVA/Tukey outcomes indicate a non-linear response to protein level, with 16% optimizing shape retention (lowest deformation) and 20% maximizing firmness, consistent with rheological stiffening at higher protein contents. These findings are in agreement with Liu et al. (2019) [31], who reported that in milk protein composite gels, printing quality improved with increasing protein concentration up to an optimal threshold (450 g/L), after which further increases (to 500 g/L) caused excessive hardness and poor elasticity, leading to surface roughness and structural imperfections.

The 16% protein formulation also ensured notable nutritional enhancement, yielding a protein content of approximately 28 g per 100 g snack on a dry matter basis (Table 4). This protein level was selected as the reference for subsequent optimization of the almond-to-yeast protein ratio.

Building upon the optimal formulation identified at 16% total protein, the ratio between almond protein and YP (100:0, 0:100, 75:25, 25:75, and 50:50 (*w*/*w*)) was varied to assess the contribution of each source and their combined effect on dough handling and printing performance (Figure 2). The physical and dimensional characteristics of the resulting snacks are summarized in Appendix A, while the corresponding statistical analyses will be provided in Appendix A. According to the one-way ANOVA and Tukey’s HSD post hoc comparisons presented in Appendix A, the almond-to-yeast protein ratio (at 16% total protein) had no significant effect on shape fidelity (*p* = 0.733), indicating comparable performance among all formulations. In contrast, significant between-group differences were observed for both deformation (*p* < 0.001) and hardness (*p* < 0.001), with multiple pairwise differences identified through post hoc analysis. These results demonstrate that while shape fidelity remained unaffected by the protein source ratio, variations in the almond-to-yeast protein composition had a pronounced impact on the mechanical properties of the printed samples.

The formulation enriched only with almond protein (100:0) showed the best shape fidelity (length = 103.71%, width = 104.13%, height = 96.44%) but also exhibited the greatest hardness value (24.03 N) among all formulations, accompanied by greater deformation (length = 5.38%, width = 3.65%, height = 8.82%). The obtained results indicate the formation of a dense and cohesive network with reduced elasticity. Similar effects were reported by Andersson (2016) [34], who observed that the addition of almond protein increased muffins’ hardness and decreased elasticity, producing dry and grainy textures. This behavior can be explained by the structural and functional properties of almond protein. It exhibits superior network-forming potential due to its emulsifying and gelation capacity, which promotes the formation of a compact and cohesive matrix [17]. Its amino acid profile is dominated by glutamic acid, arginine, aspartic acid, and leucine [35], which, together with a higher proportion of charged amino acids, enhances solubility and facilitates intermolecular bonding in aqueous systems. According to previous research [17], almond protein demonstrates strong gelling ability stabilized through both covalent (disulfide) and non-covalent (hydrogen, hydrophobic, and electrostatic) interactions. These interactions reinforce matrix cohesion, explaining the dense and rigid structure observed in almond-dominant formulations.

At a 75:25 almond-to-yeast ratio, the dough retained favorable structural integrity (length = 104.46%, width = 105.22%, height = 96.61%) and slightly reduced hardness (21.71 N), suggesting improved viscoelastic balance. While the almond protein ensured mechanical strength and matrix cohesion, partial inclusion of yeast protein introduced flexibility, reducing rigidity and improving extrusion smoothness.

Further increase in proportions of YP (25:75 ratio) led to a significant reduction in hardness (7.44 N) and slightly diminished shape fidelity (length = 104.18%, width = 105.91%, height = 95.33%). Although the dough remained extrudable and visually homogeneous, the printed structures were fragile and prone to collapse after baking, indicating insufficient matrix cohesion. The dominance of YP (0:100) further intensified these effects, resulting in the lowest hardness (3.58 N) and reduced shape fidelity (height = 92.50%), with spreading observed in printed samples.

Yeast proteins possess a relatively low sulfhydryl (–SH) group content, which limits their capacity to form disulfide cross-links, and rheological studies have shown nearly equivalent storage (G’) and loss (G″) moduli, indicative of weak gelation capacity [36]. Consequently, matrices dominated by yeast protein lack a cohesive network and exhibit reduced elasticity, poor layer adhesion, and inferior post-printing dimensional stability.

The formulation containing a 50:50 almond-to-yeast protein ratio exhibited the most balanced performance across all evaluated parameters (Figure 2). Its shape fidelity values (length = 103.29%, width = 105.06%, height = 96.83%) were closest to the target value of 100%, with low deformation (<4%) indicating precise layer deposition and high dimensional accuracy. The intermediate hardness (15.22 N), positioned between the almond-only (24.03 N) and yeast-only (3.58 N) samples, suggested an optimal balance between rigidity and flexibility. Although slightly lower in shape stability than the almond-only formulation, its favorable texture, minimal deformation, and well-defined printed layers placed it first overall in terms of print quality and structural performance.

Molecular forces governing protein–protein interactions include a combination of non-covalent interactions (electrostatic, hydrogen, hydrophobic, van der Waals, steric, and hydration repulsive forces) and covalent disulfide bonds. These interactions strongly influence protein microstructure and functional behavior, particularly solubility, emulsification, and gelation capacity [37]. When yeast and almond proteins were combined, a synergistic effect was likely achieved, leading to doughs with improved viscoelastic properties, satisfactory printability, and enhanced structural stability. The optimal performance of the 50:50 formulation can therefore be attributed to the complementary molecular characteristics of the two proteins. Almond protein provides a strong, elastic network through extensive intra- and intermolecular bonding, while yeast protein contributes hydration capacity and flexibility.

### 3.2. Rheological Properties

Rheology is the key parameter that affects the performance of the food ink during the extrusion-based 3D-printing process. Printable formulations need to have appropriate rheological properties that allow extrusion through the nozzle tip without fracture and a rapid solidification once the material has been printed to ensure the fidelity of the shape [38]. The Linear Viscoelastic Region (LVR) of the control and the previously defined optimal protein-enriched formulation was evaluated to analyze the relation among rheological characteristics and their influence on the printing process. The elastic modulus (G′, Pa), viscous modulus (G″, Pa), loss factor (tanδ = G″/G′), and complex viscosity (η*, Pas) at a frequency of 1 Hz were 12414 ± 3025 Pa, 4173 ± 396 Pa, 0.351 ± 0.079, and 742 ± 400 Pas, respectively. Throughout the frequency sweep, the general trend was detected for both G′ and G″, with the G′ value exceeding G″ (Figure 3A). This observation indicates a solid-like behavior of the formulation, with the elasticity dominating over viscosity, ensuring resistance to collapsing [39]. Furthermore, values of the loss factor less than 1 also confirm a predominantly elastic nature (Figure 3B). A comparable ratio of G″/G′, where G′ is 3–4 times higher than G″, was also reported in a study on pea protein-enriched 3D-printable dough [14]. The protein-enriched sample had higher values of loss and storage modulus than the control dough, suggesting that the addition of protein enhanced product structure [40] (Figure 3A).

The apparent viscosity, the ratio between the shear stress and shear rate, is directly affected by its composition and plays a crucial role in 3D printing [41]. The most suitable formulation should have a viscosity that is high enough to sustain its structure but low enough to enable extrusion [39]. Flow curves are presented in Figure 3C,D. An increase in shear stress and a decrease in viscosity were observed as the shear rate advanced, indicating that both the control and protein-enriched formulation showed shear-thinning behavior. Hence, these materials show a potential to be printable, as their shear-thinning nature ensures easy flow throughout deposition and optimal mechanical characteristics of the printed object once it leaves the nozzle [42].

Shear stress and shear rate data were fitted to the Power Law model (η = Kγ^(n−1)^) (Table 2), which represents a simple estimate employed to characterize the viscous flow behavior of non-Newtonian fluids [43]. K and n are Power Law parameters, where the consistency index (K) indicates the material’s viscosity, and the flow behavior index (n) describes its pseudo-plasticity behavior. K positively correlates with the viscosity and represents easier extrusion, whereas n shows an inverse trend, with values of n < 1 referring to shear-thinning behavior. As illustrated in Table 2, both formulations showed a shear-thinning behavior, where the protein-enriched sample had a slightly higher n than the control sample. Higher values of K in the case of protein-enriched formulations result from the increased viscosity due to the addition of proteins. The dough fits the Power Law model well, achieving an R^2^ of 0.970 for the control sample and R^2^ of 0.979 for the protein-enriched sample.

### 3.3. Print and End-Product Quality of 3D-Printed Snacks

The main physical and dimensional characteristics of printed snacks are presented in Table 3. The obtained results showed that the enriched formulation exhibited better fidelity in length (103.29% vs. 105.30%), width (105.06% vs. 109.01%) and height (96.83% vs. 77.78%), compared to control sample. These differences in fidelity can be directly attributed to the rheological properties of the tested formulations. Rheological data indicated that the protein- and fiber-enriched formulation demonstrated more pronounced elasticity (higher G′ value), which is critical for maintaining shape during and after printing [44]. The dominance of elastic over viscous components, along with pronounced pseudoplastic behavior, enables smooth flow through the nozzle and rapid formation of a stable structure [14].

Structural deformation after thermal processing was defined as the change in sample dimensions compared to the printed form, where positive values indicate shrinkage and negative values indicate expansion [43]. In this context, the enriched sample showed a clear advantage in terms of baking stability, with lower deformation values in width (2.31% vs. 3.75%) and height (2.98% vs. 14.02%). Although length deformation was slightly higher in the enriched sample (4.34% vs. 3.64%), it did not adversely affect the overall geometric stability.

Lastly, the baking loss represents an important quality parameter, and it is expressed as the percentage reduction in sample mass after thermal processing [45]. Baking loss was slightly lower in the enriched sample (38.16%) compared to the control (45.91%). Overall, these results indicate a high level of water loss and that the protein enrichment did not negatively impact moisture retention during baking.

These results confirm that the addition of functional components can contribute to improving printing accuracy, shape retention and stability of the three-dimensional structures, which is a key criterion for achieving high quality in 3D food printing. Since both samples were produced under identical printing conditions, the obtained results allow for a direct comparison and objective assessment of the impact of the formulation on the product behavior during and after the printing process.

### 3.4. Nutritional Characterization

Protein, carbohydrate, and fat represent the major macronutrient components of proso millet-based snacks (Table 4). Compared to the control, the enriched snack showed more than a two-fold increase in crude protein and nearly a two-fold increase in total dietary fiber. Moisture and ash content also increased, whereas fat content showed only a modest rise. In contrast, carbohydrate content decreased by approximately 25%, reflecting the reduction in proso millet flour and its replacement with protein-rich ingredients.

Proso millet is an underutilized cereal crop recognized for its high nutritional value, comprising 56–73% carbohydrates, 7–21% protein, 1–5% fat, and 2–13% crude fiber [46]. Kalinova and Moudry [47] further reported that proso millet contains an average protein content of 13%, and it is particularly rich in essential amino acids. As previously reported, among products derived from proso millet, extruded snacks demonstrated the highest protein content (11.3%) compared to other items such as muffins, couscous and porridge [48]. The incorporation of diverse plant-based proteins enables food manufacturers to enhance the nutritional quality of snack products, formulating items that qualify as “high in protein”, typically containing 10–20 g of protein per serving. Such formulations may provide up to 50% more protein than conventional animal-based counterparts, aligning with growing consumer demand for health-conscious, sustainable, and nutritionally balanced food alternatives [12].

The protein-enriched 3D-printed snack exhibited a considerably higher protein content compared to control snack, highlighting the strong impact of protein fortification on the formulation (Table 4). Previous studies have generally reported lower protein content in both conventional and 3D-printed snacks, which further emphasizes the significance of the increased protein level achieved in the fortified product. Extrusion technology has been applied to develop functional snacks by combining defatted almond cake with pearl millet, resulting in a product containing 15.45% protein [49]. In addition, the combination of milk powder and wholegrain rye flour has shown potential for producing snacks rich in both protein and dietary fiber through extrusion-based 3D printing followed by baking [50].

Recent research on extrusion-based 3D-printed snacks has predominantly focused on gluten-containing cereal matrices, such as wheat, and blends of wheat and other flours [51,52,53,54]. Gluten provides a naturally cohesive viscoelastic network that supports extrusion and enables shape retention, which explains why most available formulations rely on gluten-based systems. However, current industry trends and regulatory recommendations increasingly promote the development of gluten-free snack formulations, particularly for consumers with gluten-related disorders or those seeking allergen-reduced products [1,2]. There is still a limited number of studies using gluten-free flours in the development of 3D-printed snacks. Grain-based approaches, including those with rye flour [44], oat flour [44] and composite blends [13,38], as well as millet-based formulations [13,55], have confirmed the potential of gluten-free cereals in 3D printing. Most research on protein enrichment in gluten-free systems has relied on alternative raw materials such as chickpea and lupin [56], insect powders [57], pea protein [14], or microalgae [2]. These studies emphasize the importance of carefully adjusting ingredient ratios to achieve favorable rheological properties of the dough alongside nutritional improvement and sensory properties. This work successfully achieved substantial protein enrichment in a gluten-free matrix while maintaining good printability and post-processing stability. It highlights the use of two protein sources to enhance protein content and demonstrates the potential of proso millet as a promising gluten-free base for nutritionally improved 3D-printed snacks.

Since the product contains more than 20% of its energy value from protein, protein-enriched snacks intended for commercial distribution can be labeled as ‘high protein’ according to Regulation (EC) No. 1924/2006 [58]. Furthermore, with a fiber content of 12.17%, these snacks also meet the criteria for the “high fiber” claim. The content of insoluble dietary fiber (10.06%) in the enriched 3D-printed snack is significant compared to traditionally produced snacks, where the extrusion cooking process typically leads to a substantial reduction in insoluble dietary fiber [59,60]. High-protein and high-fiber snacks contribute to satiety and help regulate blood glucose and lipid levels, thereby supporting improved metabolic health. These properties make them particularly valuable in the dietary management of obesity and related metabolic disorders [61,62].

### 3.5. Amino Acid Profile

The amino acid profiles of the control and enriched snacks are presented in Table 5. The results indicate a substantial increase, averaging 50% or more in all amino acid contents in the enriched snacks compared to the control samples. The nutritional quality of protein is largely determined by its essential amino acid (EAA) content. Consequently, snacks enriched with almond and yeast protein contained 48% more total essential amino acids by weight than the control. Significant increases were observed in individual EAAs such as lysine, leucine, isoleucine, tryptophan, arginine, threonine, valine, phenylalanine, tyrosine, and methionine in the PRS compared to the control. The EAA content in the control snack primarily originates from proso millet and aligns closely with findings reported by Kalinova and Moudry [47], who investigated variations in amino acid content among different proso millet varieties. The non-essential amino acids detected included alanine, arginine, asparagine, cysteine, glutamic acid, glycine, proline, serine, and tyrosine. The supplementation of proso-based snacks with yeast and almond proteins led to an average increase of 63% in total non-essential amino acids. Among these, glutamic acid and aspartic acid were predominant, increasing from 4.25 to 5.54 g/100 g sample and from 1.62 to 3.40 g/100 g sample in the control and enriched snacks, respectively. Proso millet is notably rich in these amino acids [63].

Snacks prepared solely with proso millet flour exhibited the lowest lysine content, approximately 39% of the reference value from the WHO/FAO/UNU Expert Consultation (2007) [26] for adult amino acid requirements. This identified lysine as the most limiting amino acid. Lysine content increased from 0.23 g/100 g of sample in the control to 1.59 g/100 g of snacks enriched with almond and yeast protein, thereby meeting the recommended amino acid levels for reference proteins. According to the amino acid profile recommended by the FAO, almond kernel protein is considered an ideal protein due to its well-balanced amino acid composition [64]. However, the relative abundance of individual amino acids varies in the different almond kernel protein fractions, with glutamic acid being the most dominant, followed by arginine, aspartic acid and leucine. Previous research found that methionine, lysine, and threonine are the limiting amino acids in defatted almond flour [65]. Combining almond kernel protein with other protein sources can compensate for its lower lysine and methionine content. Therefore, the high lysine content in yeast protein is a critical nutritional characteristic, making yeast an excellent supplementary protein source for cereals and almonds deficient in lysine [66]. Although whole yeast (*Saccharomyces cerevisiae*) has relatively low leucine levels [67], almond protein effectively compensates for this deficiency. Consequently, the combination of yeast and almond protein isolates, when incorporated with proso millet as a base, provides a well-balanced amino acid profile with numerous potential health benefits.

Amino acid composition was further assessed by calculating the Amino Acid Score (AAS) according to the FAO/WHO/UNU [26] reference pattern for adults (Table 6). The enriched formulation showed good essential amino acid coverage, particularly for lysine, which increased substantially after enrichment. The control sample was limited in lysine (AAS ≈ 0.39), whereas the enriched snack demonstrated a significant improvement (AAS = 1.25), effectively overcoming the lysine deficiency typical of cereal-based products. In the enriched sample, the sulfur amino acids emerged as the likely limiting factor (AAS = 0.89 for Met), while most other essential amino acids were present at adequate levels (AAS > 1).

It is important to note that the AAS represents only one component of protein quality. The digestibility of protein and the effective utilization of amino acids are equally crucial for determining overall quality. The Protein Digestibility-Corrected Amino Acid Score (PDCAAS) is calculated as the product of the AAS and the percent of protein digestibility of the examined food. This approach accounts not only for amino acid composition but also for how efficiently the protein is digested and utilized in the body. For almond protein, the literature reports PDCAAS values in the range of 44.3–47.8, which are below the threshold of ~80% typically required to qualify for a “Good Source of Protein” claim on food labels. This limitation in PDCAAS largely reflects the low lysine content that is characteristic of nuts [68]. For yeast protein the PDCAAS is approximately 100%, indicating that this formulation could fully meet the indispensable amino acid requirements of older children, adolescents, and adults [69]. Our results suggest that combining almond with yeast protein can help overcome this lysine deficiency, thus improving both the amino acid profile and the potential protein quality of the final product

### 3.6. Fatty Acid Composition

The fatty acid composition, as well as the proportions of total saturated fatty acids (ΣSFA), monounsaturated fatty acids (ΣMUFA) and polyunsaturated fatty acids (ΣPUFA) in control and enriched snacks, is presented in Table 7.

The fatty acid profile mainly results from the addition of almond butter in both the control and enriched snacks, which is known to confer potential health benefits. A key reason for the health benefits of almonds is their high concentration of monounsaturated fatty acids [70]. Despite some variations, the predominant fatty acids identified in both control and enriched snacks were consistent, with oleic acid (C18:1 ω9) and linoleic acid (C18:2 ω6) being the most abundant. Regarding the total fatty acid composition of snacks, unsaturated fatty acids, specifically oleic acid and linoleic acid, accounted for approximately 54% and 28%, respectively, while the saturated fatty acids palmitic acid and stearic acid represented 7.2% and 1.8%, respectively. These findings align with previous studies conducted by Ouzir et al. [71] and Özcan et al. [72]. Similarly, a fatty acid profile analysis of almonds by Čolić et al. [73] reported oleic and linoleic acids as the dominant fatty acids, too.

The presence of trans fatty acids in salty snack products is a significant dietary concern. A study by Timić et al. [74], which examined the fatty acid composition of 58 salty snack samples from the Serbian market, established a detailed database of saturated and trans fatty acids. Their results indicated that consumption of even a single package of certain products may exceed the recommended maximum intake of 1% of daily energy from trans fatty acids. In contrast, protein-based snacks exhibited a more favorable fatty acid profile, characterized by lower levels of saturated and trans fats, suggesting that they may represent a healthier alternative to conventional snack products.

### 3.7. Texture

Food texture is a key factor influencing consumer acceptance [75]. The control sample exhibited a hardness of 12.13 N, while the protein-enriched snack showed a slightly higher value of 15.22 N (Table 3). This minor increase in hardness with higher protein content aligns with findings reported by Johansson et al. [76]. Such behavior may be attributed to the good compatibility and synergistic interactions between proteins and fibers in the dough matrix.

However, when compared with other studies utilizing gluten-free millet flour as the dough base, the obtained hardness values were significantly lower [8]. The fat content in the dough formulation used for 3D food printing plays a critical role in softening the texture of the final product. Therefore, optimizing the fat phase is essential not only for processing performance but also for sensory quality [77].

From a printing perspective, the low infill density (30%) contributed positively to the reduced hardness of the 3D-printed snack. Numerous studies have confirmed a positive correlation between infill density and snack hardness [78,79]. In addition to infill percentage, print speed and layer height also significantly affect texture. These two parameters tend to be negatively correlated with hardness: higher print speed results in less material extruded per unit time, and increased layer height reduces layer compaction and increases internal porosity, effectively decreasing hardness.

### 3.8. Sensory Analysis

A sensory analysis was performed on both the control and enriched snacks. The evaluation of sensory attributes (Figure 4) indicated that the enriched sample received higher scores for the majority of attributes compared to the control sample. The most pronounced differences were observed in flavor, smell, and aroma, where the enriched snack received significantly higher scores compared to the control (3.35 vs. 2.33, 4.25 vs. 3.15, and 3.40 vs. 2.60, respectively; *p* < 0.05). Smaller improvements were also noted in appearance (4.58 vs. 4.30) and color (3.95 vs. 3.80). In contrast, the differences in texture and crispiness were minor and did not reach statistical significance.

The overall acceptance scores indicated a clear preference for the enriched snack, with a mean score of 3.64, as compared to 3.07 for the control group. With respect to the re-consumption intention, a majority of the panelists indicated their preference for purchasing the enriched snack. Conversely, for the control snack, the trend was reversed, with a lower proportion of panelists expressing purchase preference.

The findings indicate that the addition of YP and almond protein enrichment resulted in a notable enhancement of the sensory attributes of the product, particularly in terms of flavor-related characteristics. This is in line with previous reports that protein enrichment with almond flour contributes to a pleasant, nutty flavor, which is generally preferred by consumers [34]. This trend is clearly illustrated in the radar chart (Figure 4), where the enriched sample demonstrates a broader and more balanced sensory profile compared to the control.

### 3.9. Color and Water Activity

Color is a key quality attribute that influences consumer perception and acceptance of food products. The color parameters (L*, a* and b*) are presented in Table 8. The enriched snack exhibited lower L* values (63.74) compared to the control (69.37), indicating a darker appearance. The a* value increased from 4.67 in the control to 8.12 in the enriched snack, reflecting a shift toward red tones, while b* value increased from 24.47 to 27.44, indicating yellowness. The calculated total color difference suggests a slightly perceptible color change between the two products, which can be attributed to the color of added protein ingredients and Maillard reactions occurring during post-processing [80].

Other than texture and color, moisture content is also a critical parameter influencing product quality. Water activity reflects the availability of water molecules to participate in microbial, enzymatic, or chemical reactions. Therefore, it serves as an important parameter for assessing the potential for microbial growth and the chemical stability of foods. It is well established that bacteria do not proliferate at a_w_ values below 0.80, while the threshold for mold and yeast growth is approximately 0.60 [81]. Water activity values for control and enriched snacks are 0.329 and 0.330, respectively (Table 8). These values indicate that protein enrichment had no adverse effects on this parameter. Both values are well below microbial growth threshold, indicating good microbial stability and prolonged shelf life of snacks.

### 3.10. Preliminary Laboratory-Scale Cost Benchmarking

A preliminary laboratory-scale cost benchmarking was conducted to provide an illustrative, order-of-magnitude comparison between the control and protein-enriched 3D-printed snack formulations under experimental conditions only (Table 9). The calculation included raw material costs, electricity for 3D printing and baking, and personnel costs. Based on European retail ingredient prices and late-2024 electricity rates (~0.145€ per kWh), the control formulation amounted to 1.61€ per 100 g mixture (≈0.17€ per snack), while the enriched formulation reached 2.19€ per 100 g mixture (≈0.23€ per snack).

It is important to emphasize that these values reflect small-scale experimental conditions only. In real industrial environments, cost structures are fundamentally different due to economies of scale, bulk raw material procurement, automation of labor, and continuous production systems, which can substantially alter both absolute and relative costs. Therefore, no direct conclusions regarding industrial economic feasibility can be drawn from the present data.

Nevertheless, the preliminary comparison indicates that the nutritional gain achieved through protein fortification (high protein, high fiber, improved amino acid profile) is obtained with a moderate increase in laboratory-scale production cost, supporting the formulation’s potential relevance for future pilot-scale evaluation. Despite these higher laboratory-scale costs, the enriched formulation delivers significantly greater nutritional value, supporting its relevance for future research in the field of functional and personalized nutrition. Previous studies have shown that 3D food printing holds strong potential for nutritionally tailored foods for specific population groups such as athletes, hospital patients, and seniors, indicating that protein-enriched 3D-printed snacks represent a promising direction for further pilot-scale and application-oriented investigation [82].

## 4. Conclusions

This study highlights the potential of blending proso millet flour, yeast-derived protein, and almond protein into an effective formulation enabling high-protein, fiber-rich, gluten-free dough suitable for 3D food printing.

The formulated dough, containing 16% total protein with a 50:50 almond-to-yeast protein ratio, achieves a balance between nutritional density, specifically high protein and fiber content, and physical stability, meeting the stringent requirements of 3D food printing. This combination allows for precise layer-by-layer deposition, shape retention, and post-processing texture appeal in the final snack. The findings demonstrated that the protein-enriched dough exhibited favorable rheological properties, characterized by a predominant elastic response (G′ > G″) and a shear-thinning (pseudoplastic) flow behavior. These traits enabled smooth extrusion and ensured the printed structures maintained their form both during and after the 3D printing process. Improved dimensional accuracy and minimized deformation following post-processing further emphasized the beneficial role of protein fortification in stabilizing the final product. Texture analysis showed no significant increase in hardness, maintaining consumer-acceptable texture despite the higher protein content.

Nutritional analyses showed a substantial increase in protein (28.16 g/100 g DB) and dietary fiber (12.17 g/100 g DB) in the enriched snacks. Additionally, the amino acid profile was notably improved, particularly with higher level of lysine. The fatty acid composition was also favorable, with monounsaturated and polyunsaturated fatty acids being predominant. Sensory evaluation indicated that the enriched snacks were preferred over the control, with higher scores for flavor, aroma, and overall acceptance, demonstrating that protein fortification can enhance both nutrition and consumer appeal.

Overall, this work contributes to the advancement of clean-label, functional snack development and supports the use of sustainable, plant- and microbial-based proteins in innovative food applications.

A preliminary laboratory-scale cost benchmarking indicated that the improved nutritional profile of the enriched formulation is achieved with a moderate increase in experimental production cost. However, true industrial economic feasibility, production scalability, and commercial implementation require dedicated pilot-scale trials and comprehensive techno-economic assessment, which were beyond the scope of the present proof-of-concept study.

## Figures and Tables

**Figure 1 foods-14-04308-f001:**
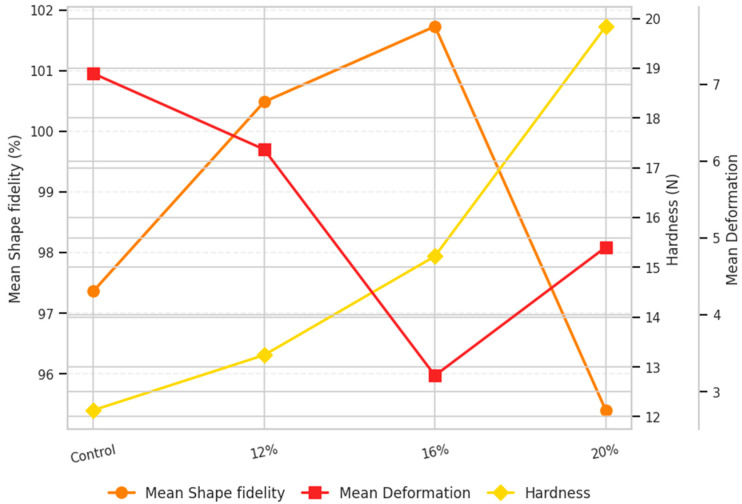
Trends in shape fidelity, deformation, and hardness of 3D-printed millet-based snacks with varying protein levels.

**Figure 2 foods-14-04308-f002:**
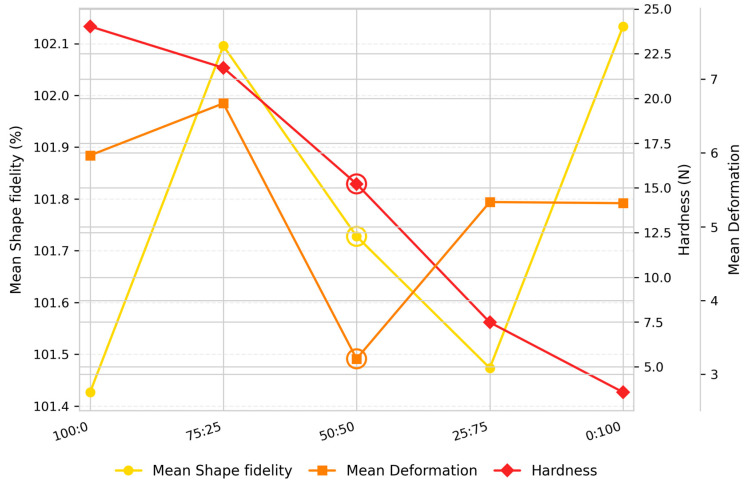
Trends in shape fidelity, deformation, and hardness of 3D-printed millet-based snacks with varying protein ratios.

**Figure 3 foods-14-04308-f003:**
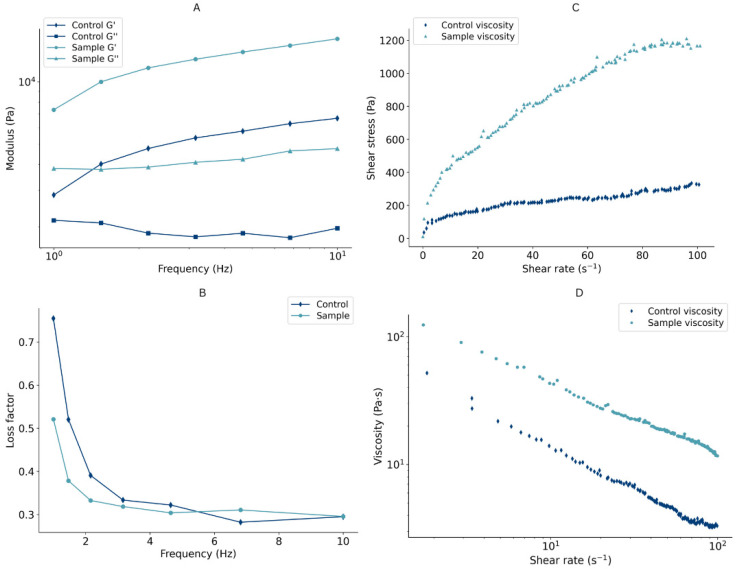
(**A**–**D**) Rheological properties of formulated dough.

**Figure 4 foods-14-04308-f004:**
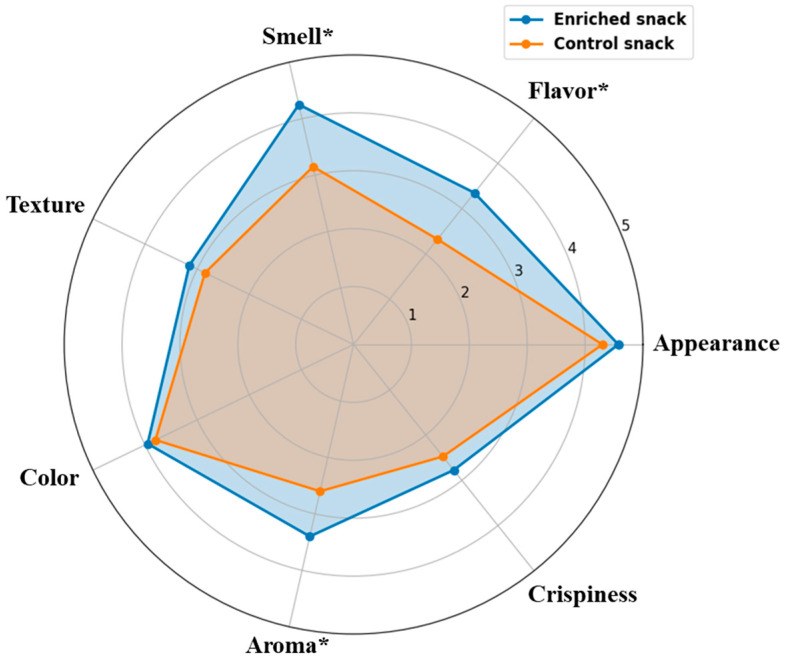
Sensory analysis results (n = 40) for control and enriched snacks. Statistically significant differences (*p* < 0.05) are indicated with an asterisk (*).

**Table 1 foods-14-04308-t001:** Control and enriched ink formulations.

Ingredient Content (%)	Control	12% Protein Enrichment	16% Protein Enrichment					20% Protein Enrichment
			100:0	75:25	50:50	25:75	0:100	
Proso millet flour	48	36	32	32	32	32	32	28
Almond protein	/	6	16	12	8	4	/	10
*Saccharomyces cerevisiae* (yeast) protein cookie flavor	/	6	/	4	8	12	16	10
Almond butter with cinnamon	11	11	11	11	11	11	11	11
Tap water	41	41	41	41	41	41	41	41

**Table 2 foods-14-04308-t002:** The model parameters of the formulated dough.

Formulations	n	K (Pa·s)	R^2^
Control	0.36 ± 0.01 ^a^	58.72 ± 1.74 ^a^	0.97
Sample	0.46 ± 0.01 ^b^	148.14 ± 3.34 ^b^	0.99

Statistically significant differences between samples (*p* < 0.05) are indicated by different letters for each parameter.

**Table 3 foods-14-04308-t003:** Physical and dimensional characteristics of control and enriched snacks.

	Weight (g)	Shape Fidelity (%)	Deformation (%)	Texture	Appearance
	*m* _0_	*m* _1_	Length	Width	Height	Length	Width	Height	Hardness(N)	Top View	Lateral View
Control snack	10.28 ± 0.14	5.56 ± 0.21	105.30 ± 0.65	109.01 ± 1.45	77.78 ± 3.55	3.64 ± 0.29	3.75 ± 0.73	14.02 ± 1.86	12.13 ± 0.52	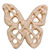	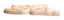
Enriched snack	10.43 ± 0.11	6.45 ± 0.07	103.29 ± 0.73	105.06 ±0.62	96.83 ± 0.88	4.34 ± 0.89	2.31 ± 0.31	2.98 ± 0.42	15.22 ± 0.77	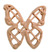	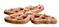

*m*_0_-weight of sample after printing, *m*_1_-weight of sample after post-processing.

**Table 4 foods-14-04308-t004:** Proximate composition of control and enriched snacks.

Nutrients (%)	Control Snack	Enriched Snack
Moisture	2.27 ± 0.41 ^a^	4.86 ± 0.54 ^b^
Ash	1.43 ± 0.30 ^a^	2.11 ± 0.32 ^a^
Total fat	12.39 ± 1.62 ^a^	14.09 ± 1.93 ^a^
Protein	13.26 ± 1.21 ^a^	28.16 ± 2.87 ^b^
Carbohydrates	72.92 ± 3.97	55.63 ± 2.56
Reducing sugars	0.74 ± 0.09 ^a^	1.38 ± 0.18 ^b^
Total dietary fiber	6.22 ± 1.25 ^a^	12.17 ± 1.87 ^b^
Insoluble dietary fiber	4.89 ± 1.03 ^a^	10.06 ± 1.94 ^b^
Soluble dietary fiber	1.33 ± 0.13 ^a^	2.11 ± 0.58 ^b^

Values are presented as means ± standard deviation (n = 3). Different letters within the same row indicate statistically significant differences between samples (*p* < 0.05). Carbohydrate content was determined by difference. All results are reported on a dry matter basis (g/100 g DB).

**Table 5 foods-14-04308-t005:** Total amino acid (TAA) composition of the control and enriched snacks (g/100 g sample).

Amino Acids	Control Snack	Enriched Snack
Aspartic acid	1.62 ± 0.25 ^a^	3.40 ± 0.87 ^b^
Arginine	0.74 ± 0.15 ^a^	1.86 ± 0.98 ^a^
Serine	1.11 ± 0.06 ^a^	1.77 ± 0.08 ^b^
Glutamic acid	4.25 ± 1.07 ^a^	5.54 ± 1.14 ^a^
Proline	0.78 ± 0.05 ^a^	1.04 ± 0.07 ^b^
Glycine	0.67 ± 0.04 ^a^	1.42 ± 0.09 ^b^
Alanine	1.53 ± 0.09 ^a^	1.90 ± 1.00 ^a^
ΣNEAA	10.70 ± 1.98 ^a^	16.93 ± 2.44 ^b^
Valine	0.89 ± 0.08 ^a^	1.72 ± 0.09 ^b^
Methionine	0.19 ± 0.01 ^a^	0.35 ± 0.02 ^b^
Isoleucine	0.61 ± 0.01 ^a^	1.32 ± 0.10 ^b^
Leucine	1.61 ± 0.08 ^a^	2.36 ± 0.10 ^b^
Tyrosine	0.36 ± 0.02 ^a^	0.89 ± 0.07 ^b^
Phenylalanine	0.89 ± 0.09 ^a^	1.62 ± 0.09 ^b^
Lysine	0.23 ± 0.01 ^a^	1.59 ± 0.40 ^b^
Histidine	0.35 ± 0.03 ^a^	0.66 ± 0.05 ^b^
Threonine	0.54 ± 0.04 ^a^	1.27 ± 0.09 ^b^
ΣEAA	5.67 ± 1.25 ^a^	11.78 ± 1.98 ^b^
Total amino acids (TAA)	16.37 ± 2.25 ^a^	28.70 ± 3.05 ^b^

ΣNEAA—non-essential amino acids, ΣEAA—essential amino acids. Values are presented as means ± standard deviation (n = 3). Different letters within the same row indicate statistically significant differences between samples (*p* < 0.05). All results are reported on a total protein content basis.

**Table 6 foods-14-04308-t006:** Amino acid scores (AASs) of control and enriched snacks.

Essential AA	Control (mg/g Protein)	Enriched (mg/g Protein)	FAO (mg/g Protein)	AAS Control	AAS Enriched
Histidine	26.15 ± 2.65 ^b^	23.44 ± 1.45 ^a^	15	1.74 ± 0.18 ^b^	1.56 ± 0.12 ^a^
Isoleucine	46.00 ± 0.76 ^a^	46.88 ±3.56 ^a^	30	1.53 ± 0.03 ^a^	1.56 ± 0.12 ^a^
Leucine	121.42 ± 6.04 ^b^	83.09 ± 4.65 ^a^	59	2.06 ± 0.11 ^b^	1.41 ± 0.08 ^a^
Lysine	17.35 ± 0.76 ^a^	56.46 ± 14.21 ^b^	45	0.39 ± 0.02 ^a^	1.25 ± 0.32 ^b^
Methionine	14.32 ± 0.76 ^b^	12.43 ± 0.71 ^a^	16	0.89 ± 0.05 ^b^	0.77 ± 0.05 ^a^
Phenylalanine + tyrosine	94.27 ± 8.30 ^a^	89.13 ± 5.69 ^a^	38	2.48 ± 0.22 ^a^	2.34 ± 0.15 ^a^
Threonine	40.72 ± 3.02 ^a^	45.10 ± 3.20 ^a^	23	1.77 ± 0.13 ^a^	1.96 ± 0.14 ^a^
Valine	67.12 ± 6.03 ^a^	60.41 ± 2.27 ^a^	39	1.72 ± 0.16 ^a^	1.55 ± 0.06 ^a^

AA—amino acid, AAS—amino acid score. Values are presented as means ± standard deviation (n = 3). Different letters within the same row indicate statistically significant differences between samples (*p* < 0.05).

**Table 7 foods-14-04308-t007:** The fatty acid composition of control and enriched snacks.

Fatty Acid (%)	Control Snack	Enriched Snack
C8:0	0.6 ± 0.01 ^b^	0.5 ± 0.01 ^a^
C10:0	0.5 ± 0.02 ^b^	0.4 ± 0.01 ^a^
C12:0	4.2 ± 0.06 ^b^	3.4 ± 0.01 ^a^
C14:0	1.7 ± 0.04 ^b^	1.4 ± 0.01 ^a^
C16:0	7.1 ± 0.17 ^a^	7.4 ± 0.15 ^b^
C17:0	n.d.	0.1 ± 0.18
C18:0	1.7 ± 0.20 ^a^	1.9 ± 0.18 ^a^
C20:0	0.2 ± 0.03 ^a^	0.2 ± 0.21 ^a^
C22:0	0.1 ± 0.03 ^a^	0.1 ± 0.21 ^a^
C24:0	0.1 ± 0.03 ^a^	0.1 ± 0.21 ^a^
ΣSFA	16.2 ± 0.07 ^b^	15.5 ± 0.05 ^a^
C16:1	0.3 ± 0.01 ^a^	3.0 ± 0.03 ^b^
C18:1n9c	54.6 ± 0.48 ^a^	54.1 ± 0.51 ^a^
ΣMUFA	54.9 ± 3.17 ^a^	57.1 ± 3.09 ^a^
C18:2n6c	28.3 ± 1.84 ^a^	27.0 ± 1.88 ^a^
C18:3n3	0.3 ± 0.04 ^b^	0.2 ± 0.03 ^a^
C20:3n3	0.1 ± 0.01 ^a^	0.1 ± 0.02 ^a^
ΣPUFA	28.7 ± 2.61 ^a^	27.3 ± 2.55 ^a^

ΣSFA—saturated fatty acids; ΣMUFA—monounsaturated fatty acids; ΣPUFA—polyunsaturated fatty acids. Values are presented as means ± standard deviation (n = 3). Different letters within the same row indicate statistically significant differences between samples (*p* < 0.05). All results are reported on a total fat content basis.

**Table 8 foods-14-04308-t008:** Water activity, color parameters (L*, a* and b*) and total color difference (ΔE) values of post-processed control and enriched snack.

Sample	a_w_	L*	a*	b*	ΔE	Color
Control snack	0.329 ± 0.003 ^a^	69.37 ± 0.97 ^b^	4.67 ± 0.19 ^b^	24.47 ± 0.53 ^b^	-	
Enriched snack	0.330 ± 0.001 ^a^	63.74 ± 1.54 ^a^	8.12 ± 0.47 ^a^	27.44 ± 0.53 ^a^	7.27 ± 1.06	

Values are presented as means ± standard deviation (n = 3). Different letters within the same row indicate statistically significant differences between samples (*p* < 0.05).

**Table 9 foods-14-04308-t009:** Ingredient and process cost breakdown for 3D-printed snack formulations.

Ingredient/Process	Control Ink (%)	Enriched Ink (%)	Commercial Price	Cost per 100 g Mixture—Control (≈9.5 Snacks)	Cost per 100 g Mixture—Enriched (≈9.5 Snacks)
Proso millet flour	48	32	1.54€/500 g	0.15€	0.10€
Almond protein	–	8	8.35€/300 g	–	0.22€
Saccharomyces cerevisiae (yeast) protein cookie flavor	–	8	25.56€/500 g	–	0.41€
Almond butter with cinnamon	11	11	5.44€/170 g	0.35€	0.35€
Tap water	41	41	–	–	–
Electricity (3D printing, 3 min ≈ 0.05 kWh)	–	–	0.145€/kWh	0.01€	0.01€
Electricity (baking, 10 portions, 15 min ≈ 0.25 kWh/10)	–	–	0.145€/kWh	0.00€ (≈0.004€)	0.00€ (≈0.004€)
Personnel cost (5 min @€2100/month)	–	–	2100€/month	1.10€	1.10€
Total cost per 100 g mixture				1.61€ (≈0.17€ per snack)	2.19€ (≈0.23€ per snack)

Note: Monthly wage assumed at 2100€ gross, 160 h/month → ≈ 0.22€ per minute → 5 min ≈ 1.10–1.33€ (rounded depending on working hour basis).

## Data Availability

The original contributions presented in the study are included in the article/Appendix A; further inquiries can be directed to the corresponding author.

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
