# Peer review of "Protein Fortification of Millet-Based Gluten-Free Snacks Designed for 3D Printing"

_foods, 2025, doi:10.3390/foods14244308_

Round 1

Reviewer 1 Report (Previous Reviewer 1)

Comments and Suggestions for Authors

The manuscript has been improved, but still requires further improvements. These are detailed below.

  1. The expanded experimental design is adequate, but why was a formal statistical mixture design not incorporated? Please explicitly mention what type of design you used.
  2. The trends described for protein levels and mixtures remain descriptive, lacking robust comparative statistical analysis (e.g., multivariate ANOVA or interaction terms).
  3. The discussion section does not sufficiently delve into the molecular mechanisms that underlie the rheological behavior or protein synergy.
  4. The selection of the optimal level (16% and 50:50) depends on subjective criteria, without mathematical validation (e.g., response surface modeling).
  5. No validation of storage stability is included, which is essential for 3D-printed snacks.
  6. The sensory analysis is fundamental; the sample of 40 panelists does not provide robust sensory validation and lacks multivariate analysis.
  7. The cost assessment is preliminary and does not allow for determining actual industrial feasibility.
  8. Excessive claims about industrial applicability persist without sufficient experimental support.
  9. Comparison with recent studies remains limited in critical depth, especially in 3D printing technologies applied to snacks.
  10. Advanced thermal and structural parameters (TG, DSC, SEM) are not evaluated, which would better support conclusions about structure.
  11. 3D printing is well described technically, but there is no reproducibility analysis, only three replicates per formulation.

Author Response

Reviewer 2 Report (Previous Reviewer 2)

Comments and Suggestions for Authors

Dear authors,

the manuscript is significantly improved.  The used methodology for enrichments is now clear.

Round 2

Reviewer 1 Report (Previous Reviewer 1)

Comments and Suggestions for Authors

Accepted in the present form.

This manuscript is a resubmission of an earlier submission. The following is a list of the peer review reports and author responses from that submission.

Round 1

Reviewer 1 Report

Comments and Suggestions for Authors

Major Observations
1. The experimental design of the study should be improved; in its current format, there are only two formulations (control and enriched). The standards of the journal Foods are high, so the design should be improved (for example, with a mixture design or others, and the incorporation of more treatments). Doing so would allow clear trends to be observed, dose-response to be evaluated, and optimal protein levels to be determined. With only one level of enrichment, the strength of the conclusions is reduced. 
2. The manuscript proposes, as an innovative contribution, the replacement of synthetic gums with natural proteins. However, no comparison group with these gums is presented, which prevents the objective demonstration of the advantage of the proposed approach.
3. The manuscript does not demonstrate how it differs substantially from previous work with other protein sources. The discussion should delve deeper into the novelty and molecular mechanisms that explain the better performance of this combination.
4. No analyses of storage stability, sensory evaluation, cost evaluation, or possible industrial scalability are included. The absence of these reduces the potential impact of the work.
5. The results section lacks a thorough critical analysis and comprehensive comparison with recent studies, highlighting what this study uniquely contributes.
6. The conclusions state that the product is viable and promising, but with such a limited experimental design, these claims lack sufficient support. The experimental basis needs to be expanded before industrial or commercial applications can be recommended.

Reviewer 2 Report

Comments and Suggestions for Authors

Dear Authors,

The manuscript is well written and very interesting. In my opinion, it addresses some of the most intriguing aspects of innovative transformation processes, such as 3D printing combined with the nutritional enhancement of gluten-free ready-to-eat snacks.

Please find below some revision suggestions:

General comment: Many data points appear redundant, as they are reported both in the text and in the tables. I suggest including only the percentage increases or decreases relative to the control in the text.

Paragraph 2.2: The authors should clarify the rationale behind the chosen enrichment levels (8% almond protein and 8% S. cerevisiae protein). Additionally, please explain the differences in almond butter content between the control and enriched formulations, as well as the variation in added water.

Paragraph 2.9: Please describe the method used for the extraction and preparation of fatty acids.

Paragraph 2.11: Given that the butterfly shape and surface are not homogeneous, the authors should explain how color assessment was performed.

Paragraph 3.3, line 312: The total dietary fiber content differs from the values reported in Table 4. Please correct this inconsistency.

Round 2

Reviewer 1 Report

Comments and Suggestions for Authors

Despite the corrections incorporated, the manuscript retains significant methodological and conceptual limitations. The experimental design remains too restrictive, preventing the identification of clear trends and the adequate evaluation of responses. The novelty of the study remains at a preliminary level, without experimental evidence to support it, and the additional analyses are still insufficient. These weaknesses affect the robustness and practical applicability of the results; therefore, I recommend rejection, encouraging the authors to reconsider the experimental design and strengthen the validation of their contribution.